

# Double lives: transfer of fungal endophytes from leaves to woody substrates

Aaron Nelson*, Roo Vandegrift*, George C. Carroll and Bitty A. Roy

Institute of Ecology and Evolution, University of Oregon, Eugene, OR, USA
* These authors contributed equally to this work.

## ABSTRACT

Fungal endophytes are a ubiquitous feature of plants, yet for many fungi the benefits of endophytism are still unknown. The Foraging Ascomycete (FA) hypothesis proposes that saprotrophic fungi can utilize leaves both as dispersal vehicles and as resource havens during times of scarcity. The presence of saprotrophs in leaf endophyte communities has been previously observed but their ability to transfer to non-foliar saprobic substrates has not been well investigated. To assess this ability, we conducted a culture study by placing surface-sterilized leaves from a single tropical angiosperm tree (*Nectandra lineatifolia*) directly onto sterile wood fragments and incubating them for 6 weeks. Fungi from the wood were subsequently isolated in culture and identified to the genus level by ITS sequences or morphology. Four-hundred and seventy-seven fungal isolates comprising 24 taxa were cultured from the wood. Of these, 70.8% of taxa (82.3% of isolates) belong to saprotrophic genera according to the FUNGuild database. Furthermore, 27% of OTUs (6% of isolates) were basidiomycetes, an unusually high proportion compared to typical endophyte communities. *Xylaria flabelliformis*, although absent in our original isolations, formed anamorphic fruiting structures on the woody substrates. We introduce the term *viaphyte* (literally, "by way of plant") to refer to fungi that undergo an interim stage as leaf endophytes and, after leaf senescence, colonize other woody substrates via hyphal growth. Our results support the FA hypothesis and suggest that viaphytism may play a significant role in fungal dispersal.

## INTRODUCTION

Endophytes are symptomless endosymbionts of living plants (*Stone, Bacon & White, 2000*) and are ubiquitously present in terrestrial plant tissues worldwide (*Arnold & Lutzoni, 2007*). Virtually every plant genus surveyed to date has documented several to hundreds of species of fungal endophytes per individual, and a single plant species may host thousands of these symbionts across its entire range (*Martins et al., 2016*; *Barge et al., 2019*). Although variable, the effects of endophytes on host plants have attracted considerable attention (*Carroll, 1988*; *Rodriguez et al., 2009*); yet, the potential benefit of endophytic life histories for the *fungal* partners is less well explored.

The question of why fungi may adopt endophytic lifestyles has garnered a variety of hypotheses. In particular, a number of authors have hypothesized that endophytes may be

Corresponding author
Roo Vandegrift, awv@uoregon.edu

latent saprotrophs that benefit from being the first to colonize plant tissues after senescence or death of the host (*Promputtha et al., 2007*; *Parfitt et al., 2010*; *Porras-Alfaro & Bayman, 2011*; *Szink et al., 2016*), a phenomenon known as priority effects (*Chase, 2003*; *Osono, 2006*). Studies that sampled living and decomposing leaves from the same plant individuals have observed the majority of foliar endophytes can persist in the litter layer as decomposers (*Osono, 2006*; *U'Ren & Arnold, 2016*), especially in the early stages of litter decomposition, when litter contains a higher availability of simple sugars and other easily degradable compounds (*Carroll & Petrini, 1983*; *Voříšková & Baldrian, 2013*). Endophytes observed to persist into the late stages of litter decomposition (*Peršoh et al., 2013*) often have demonstrated an ability to degrade more complex substrates, such as lignin, which supports the hypothesis that some fungi with an endophytic life stage may also play a role during later stages of litter decay (*Osono & Takeda, 1999*). Although the majority of studies have focused on foliar endophytes, *Parfitt et al. (2010)* suggest that most, if not all, trees carry sapwood endophytes with the potential to degrade the woody tissues of their host when environmental and biological conditions are conducive to decay. In contrast, other studies have suggested endophytes are primarily mutualists, with their fitness directly tied to that of their hosts. This is exemplified best by clavicipitaceous grass endophytes, which benefit from direct vertical transmission to their hosts' offspring (*Clay, 1988*; *Hodgson et al., 2014*). Finally, it has been hypothesized that endophytes may be latent pathogens waiting to exploit a weakened state of their host (*Carroll, 1988*; *Slippers & Wingfield, 2007*). However, the vast majority of observed endophytic fungi do not fit neatly into one of these categories and may in fact be capable of a variety of context-dependent interactions with their hosts (i.e., endophytic continuum; *Schulz & Boyle, 2005*).

Regardless of ecological mode, the evolutionary benefits of endophytic leaf colonization for species that do not form fruiting bodies on leaves remains obscure. For instance, a number of genotypes closely related to wood decomposers have been found to also inhabit living leaves as endophytes (*Promputtha et al., 2007*), yet these taxa have not been observed to also form fruiting bodies on leaves. Thus, it has been proposed that endophytic colonization may represent an evolutionary "dead-end" (i.e., saprotrophs found as endophytes are unlikely to reproduce from leaves). This idea appears logical since most endophyte infections in living leaves remain localized, occupying only one or a few host plant cells (*Carroll, 1988*; *Bayman et al., 1998*; *Arnold & Lutzoni, 2007*), and endophytes do not usually colonize woody stems from the leaves where the infection could result in fruiting body formation (*Sun et al., 2012*; *Tateno et al., 2015*; *Thomas et al., 2019*). Yet, the colonization of live plant tissues requires specialized chemical and physical systems (*Kusari, Hertweck & Spiteller, 2012*) and the construction of such cellular mechanisms during development, along with propagule loss, incurs evolutionary costs that are unaccompanied by benefits if endophytism is truly a 'dead end' for these fungi.

One possible explanation for this discrepancy is the Foraging Ascomycete (FA) hypothesis (*Carroll, 1999*; *Thomas et al., 2016*, *2019*; *Thomas, Vandegrift & Roy, 2020*), which proposes that the function of leaf endophytism for some fungi may be to increase dispersal to other substrates by helping to bridge spatiotemporal gaps in preferred

substrate. While some saprotrophic endophytes can fruit directly from fallen leaves (*Sherwood-Pike, Stone & Carroll, 1986*; *Osono, 2006*; *Peršoh et al., 2013*), the FA hypothesis proposes that after leaves senesce and fall, leaf endophytes are capable of transferring to other substrates in their environment that are separate from their original endophytic hosts. Thus, during times of suboptimal environmental conditions, endophytes may have an increased likelihood of survival compared to spores or saprobic mycelia because the highly buffered environment of living leaves, which can provide a source of nutrients regardless of surrounding environmental conditions (*Thomas et al., 2016*). We hypothesize that the ability of spores to colonize living leaves is essentially a form of evolutionary bet-hedging that "reduces the temporal variance in fitness at the expense of a lowered arithmetic mean fitness" (*Ripa, Olofsson & Jonzén, 2010*). Direct spore dispersal by itself may result in a higher mean success rate in colonizing substrates suitable for fruiting body production, but success will be highly contingent on suitable environmental conditions (*Thomas, Vandegrift & Roy, 2020*). Thus, when a subset of spores from each sporulation event colonize leaves as endophytes, a species can decrease the variance of dispersal success (*Thomas et al., 2016*).

To encompass the processes described by the FA hypotheses, we introduce the new term *viaphyte* to refer to fungi that undergo these lifestyle shifts: the subset of endophytic fungi that are primarily saprotrophic, but which also occur as leaf endophytes and are capable of dispersal from their endophytic hosts to other substrates following leaf senescence. We create this term because (1) referring to such fungi as "foragers" is vague and leads to confusion, and (2) referring to them as "foraging ascomycetes" (or "FA utilizing fungi" and other such permutations) is inaccurate as endophytes in the Basidiomycota are likely to utilize this dispersal strategy as well (*Thomas, Vandegrift & Roy, 2020*). "Viaphyte" joins the word *via*—defined as "travelling through a place en route to a destination"—with the suffix, *phyte*, which denotes a plant. In this study, we use the term specifically to refer to fungi that display the ability to directly transfer from an endophytic state (inhabiting living leaf tissue, necessarily biotrophic) to a free-living state (inhabiting a dead woody substrate, necessarily saprotrophic) though hyphal growth.

While viaphytism is superficially similar to latent saprotrophism, it is a distinct and more complex process. Latent saprotrophy presupposes that the purpose of a fungus being present as an endophyte is to consume the tissue of its host after senescence. The idea that endophytism may be a *vehicle*, rather than an end destination, is a distinct concept. As such, the use of the term "viaphyte" helps to clarify this distinction and avoid confusion as the literature around these topics evolves.

For the FA hypothesis to be feasible (i.e., for viaphytism to occur) it must be shown that transfer from living leaves to another substrate is possible. *Thomas et al. (2016)* observed such transfer, but that study was restricted to a single fungal genus, *Xylaria*, and it is unclear how prevalent this ability is among fungal endophytes of other taxonomic groups. Here, we conducted a survey of the viaphytic abilities of endophytes present in leaves of the tropical tree, *Nectandra lineatifolia* (Ruiz & Pav.) Mez, as the tropics represent a hotspot for endophyte diversity (*Arnold & Lutzoni, 2007*). We also assessed the overall diversity of observed viaphytes and the presumed ecological roles of each isolated viaphytic

fungus. Leaf endophytes are hyperdiverse and have a wide taxonomic breadth (*Arnold et al., 2000*; *Bazzicalupo, Bálint & Schmitt, 2013*; *Thomas et al., 2019*). As a subset of the endophytic community, we expected that viaphytes would also represent a wide taxonomic breadth. Despite the fact that source communities were likely to harbor many biotrophs capable of facultative saprotrophy, based on the framework of the FA hypothesis we hypothesized that the majority of viaphytes isolated would be taxa whose primary nutritional mode is saprotrophy.

## MATERIALS AND METHODS

### Culture methods

Twelve evergreen leaves of a randomly selected tree (Lauraceae; *N. lineatifolia* (Ruiz & Pav.) Mez) were collected in an Ecuadorian cloud forest. The tree was within Reserva Los Cedros, which is on the western slope of the Andes in northwestern Ecuador (00°18031.000 N, 78°46044.600 W), at 1,200 m above sea level. Eight 2-cm$^2$ sections were cut from each leaf and surface-sterilized by successive immersion in 70% ethanol for 1 min, 5% sodium hypochlorite (equivalent to full strength bleach) for two min, then rinsed in sterile water. The leaf sections were placed onto twice-autoclaved white birch (*Betula papyrifera* Marshall) tongue depressors (Puritan, Guilford, ME, USA) as a standardized angiosperm woody substrate. The sections from each leaf were split between two tongue depressors (four sections each) resulting in a total of 24 tongue depressors. These were incubated in three 95% EtOH-sterilized Ziploc storage boxes (eight in each box) at the field station in ambient temperature for 6 weeks. Each box contained an open container of twice-autoclaved water to maintain humidity. The incubation period provided opportunity for the endophytic fungi in the leaves to colonize the wood. After incubation, the sticks were placed into airtight, sterile bags and brought to the University of Oregon.

Fungal cultures were isolated from the inoculated wood by breaking 15 small fragments (~5 mm$^2$ each) of wood from each tongue depressor using flame-sterilized tools and dispersing them evenly among five 100 mm water agar plates. The ends of growing hyphae were excised from the agar using a dissecting microscope and a scalpel and transferred onto nutrient plates (MEA, 2% maltose) over a 2-month period. Cultures were also made from several fruiting structures that grew directly from the birch substrate fragments. After a growth period of seven or more days the isolates were grouped into morphotypes (*Lacap, Hyde & Liew, 2003*) at the genus level based on macro-and microscopic features.

All field work was done with the approval of the Ecuadorian Ministry of the Environment (Ministerio del Ambiente de Ecuador, Permit No. 03-2011-IC-FLO-DPAI/MA).

### Identification of viaphytes

A single representative of each morphotype was subcultured in liquid media (2% malt extract) for DNA extraction using the Qiagen DNeasy Plant kit following the manufacturer's instructions, and the ITS region (the standard "barcode" locus for fungi; *Schoch et al., 2012*) was amplified using the fungal-specific primer set ITS1F (5′-CTTGGTCATTTAGAGG AAGTAA-3′) and ITS4 (5′-TCCTCCGCTTATTGATATGC-3′) (*White et al., 1990*), or in cases where those primers were ineffective, isolates were amplified with ITS5

(5′-GGAAGTAAAAGTCGTAACAAGG-3′) and LR3 (5′-CCGTGTTTCAAGACGGG-3′) primers. DNA amplification was conducted with 12.5-μL reaction volumes (2.5 μL of template, 6.25 μL of Sigma Aldrich JumpstartTM Taq ReadymixTM, 2.75 μL sterile water, 0.5 μL 25 mM MgCl$_2$ and 0.25 μL of each primer at 10 μM). PCR amplification was performed with an MJ Research PTC-200 DNA Engine thermal cycler under the following parameters: initial denaturation at 95 °C for 2 min, five cycles of denaturation at 95 °C for 30 s, annealing at 60 °C for 30 s, and extension at 72 °C for 1 min; followed by 25 cycles of denaturation of 95 °C for 30 s, annealing at 55 °C for 30 s, and extension at 72 °C for 1 min; a final extension at 72 °C for 10 min, and a final step of indefinite duration at 4 °C. PCR products were visualized on a 1% agarose gel. Samples were then frozen until shipping for sequencing at Functional Biosciences, Inc (Madison, WI, USA) on ABI 3730xl instruments using Big Dye V3.1. ITS amplicons were sequenced bi-directionally, then assembled into contigs, and manually edited in Geneious (v6.0.3; Biomatters Limited, Auckland, New Zealand) to remove priming sites and resolve mismatches. The consensus sequences were then compared to published sequences in the UNITE database (v8.0; *Kõljalg et al., 2013*) using the *assign_taxonomy.py* function from the Quantitative Insights into Microbial Ecology pipeline (*Caporaso et al., 2010*). Taxa that returned species assignments as "unidentified" were further examined using BLAST against the NCBI *nr* database. Taxonomic identities were assigned at genus level and lower if the hit with the lowest E-Value had greater than 97% sequence identity across the entire ITS region. Sequences whose hits did not match these criteria were categorized as "unidentified". Putative *Xylaria* species were compared to our database of ITS sequences generated from authenticated material within that genus at the same site (*Thomas et al., 2016*) and assigned to a taxon if sequences had greater than 98% sequence identity. Taxa with greater than 99% sequence identity were assumed to be the same taxon (i.e., OTU). All taxa with identical assignments by UNITE met this criterion.

Functional guilds were assigned to each genus by using the FUNGuild online tool (*Nguyen et al., 2016*), which assigns functional information to taxa in DNA datasets. If functional guilds were not available in FUNGuild, they were determined based on the literature wherever possible (Table S3).

## Statistical methods

Species richness per leaf was estimated using Chao2 and Jacknife1 estimators (*Burnham & Overton, 1978*; *Chao, 1984*; *Colwell & Coddington, 1994*). Diversity was estimated between all leaves, within leaves, and within boxes using Shannon's index (log base *e* was used; *Shannon, 1948*) and Simpson's index (1-*D*; *Simpson, 1949*), and community structure was visualized using non-metric multidimensional scaling and differences assessed with permutational multivariate analysis of variance (PerMANOVA). Data were analyzed using R Statistical Software, v. 3.1.0 (*R Core Team, 2014*), including the *vegan* package (*Oksanen et al., 2013*).

All scripts, data tables, and raw data (morphotype counts and sequence chromatograms) is available via an open FigShare repository (*Nelson et al., 2019*). Edited sequences have been uploaded to GenBank (accession numbers provided in Table S1).

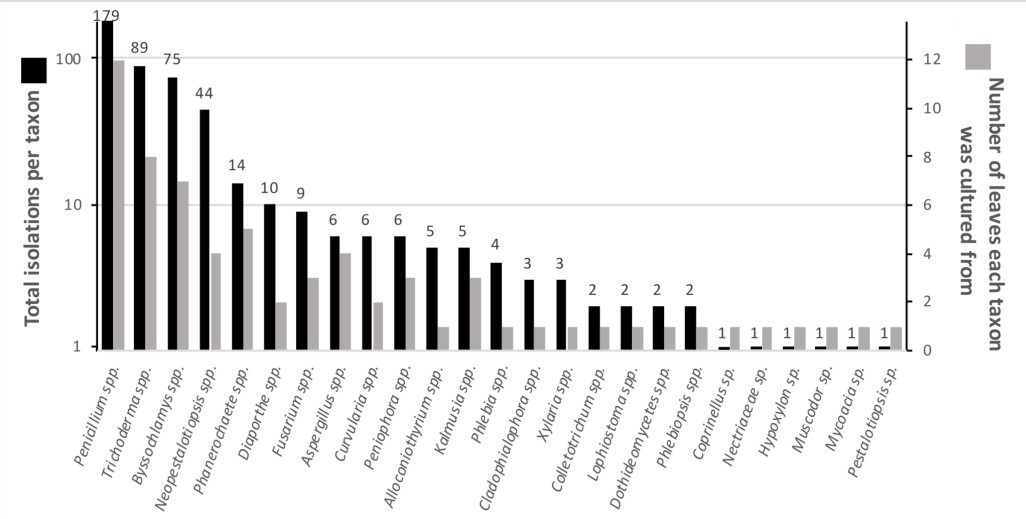

**Figure 1 Summary of identified fungal endophytes that transferred from host leaves into a woody substrate.** From 12 leaves, 25 taxa transferred to wood and were subsequently isolated. Of a total of 472 identified isolates, 82% were represented by the four most common taxa. The total isolates per taxa roughly corresponds to the number of leaves they were isolated from. The numbers on the bars specify the number of cultures per taxon. (Note: the left axis is on a logarithmic scale) five isolates remained unidentified and are not included in the figure.   

# RESULTS

## Diversity and abundance of viaphytes

Numerous endophytes from surface-sterilized leaves of *N. lineatifolia* successfully colonized the wood substrate: 477 fungal cultures were isolated after making the initial transfer from leaves to wood. Isolates were grouped into 64 morphotypes, 62 of which were successfully identified to genus (59 by DNA, three by morphology; Table S1). DNA identification resulted in the consolidation of the morphotypes into 24 unique taxa at the genus level (Table S2). The number of isolates for each taxon varied widely, such that 57% of the isolates were represented by just two genera (i.e., *Trichoderma* and *Penicillium*), and seven of the taxa were isolated only a single time (Fig. 1). In addition to hyphal growth from the wood substrates, anamorphic fruiting structures were observed growing out of five stick fragments originating from two leaves (Fig. S1). These isolates were identified as *Xylaria flabelliformis* (Schwein.) Berk. & M.A. Curtis using DNA extracted from stromatic tissues. Including *X. flabelliformis*, we observed a total of 24 viaphytic taxa, which were identified to the genus level (Fig. 1). Additionally, we observed that the majority of the woody substrate fragments displayed a dramatic decrease in substrate volume that may be explained by high levels of cell wall degrading enzymes typical of white-rot fungi. However, we did not attempt to determine which taxa were responsible for this dramatic reduction in volume.

The species accumulation curve did not reach a saturation point, suggesting that the full richness of viaphytes from these leaves was not isolated (Fig. 2). Estimates of actual species richness ranged from 36.5 (first order jackknife, SE = 4.1) to 42.3 (chao2, SE = 13.8).

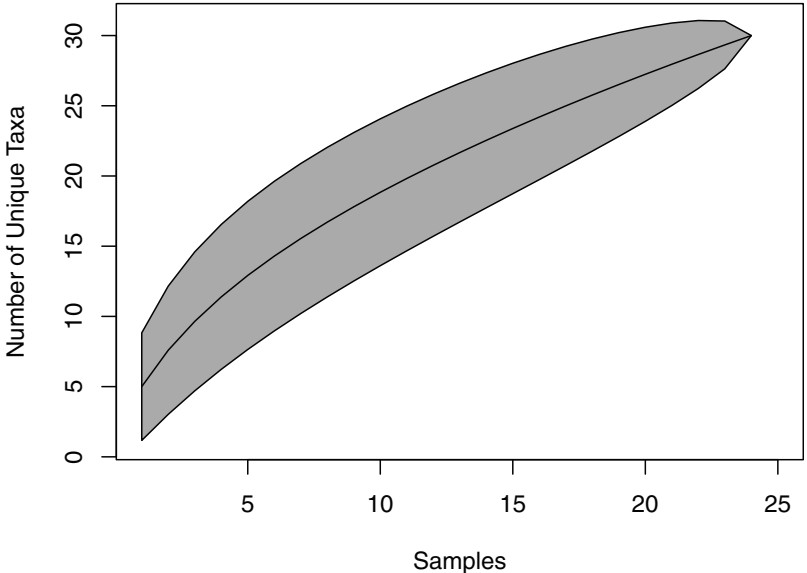

**Figure 2 Species accumulation curve for viaphytes.** The culturing did not achieve a saturation of culturable viaphytic taxa. 

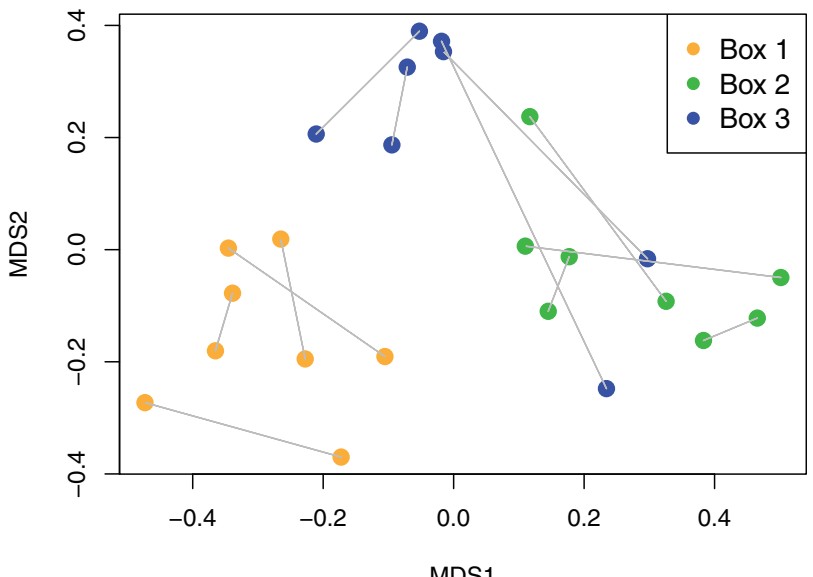

**Figure 3 Non-metric Multidimensional Scaling (NMDS) plot of viaphyte communities.** Each point represents an individual birch tongue depressor; lines connect sticks that were inoculated with the same leaf; color indicates inoculation box. 

Viaphyte communities within incubation boxes were more similar to each other than to communities from other boxes (PerMANOVA: $F_{1,23} = 6.34$, $p = 0.001$), whereas communities from sticks that were inoculated by the same leaves were not more similar to each other than to sticks inoculated from different leaves (PerMANOVA: $F_{1,23} = 1.04$, $p = 0.404$; Fig. 3). Isolates representing the four most common taxa were concentrated in common boxes, with 100% of *Neopestalotiopsis foedans* in Box 1 (44 total isolates across all

boxes), 96% of *Paecilomyces formosus* in Box 1 (75 total isolates), 87% of *Trichoderma spp.* in Box 2 (89 total isolates), and 61% of *Penicillium spp.* in Box 3 (179 total isolates).

## Taxonomic distribution

The higher order taxonomic ranks in our samples included two phyla, five classes, 12 orders and 19 families (Table S2). Although Ascomycota was the dominant phylum, both in terms of number of taxa and total number of isolates (73% and 94%, respectively), isolates of Basidiomycota also were obtained in culture. Among Ascomycota fungi, Sordariomycetes were the most common class in terms of number of taxa (38.4% of total taxa), whereas fungi in the Eurotiomycetes, driven by the frequency of *Penicillium* spp., represented more than half of the isolates (55.7%). At the ordinal level, the most common orders among all taxa were Xylariales (Sordariomycetes, Ascomycota) and Polyporales (Basidiomycta) (each representing 19.2% of all taxa). Isolates of Eurotiales (Eurotiomycetes, Ascomycota), again driven by *Penicillium* spp., represented the most isolates (55.1% of all isolates).

## Functional guilds

The FUNGuild database contained putative functional guilds for all but two of the genera we isolated as viaphytes. The first unassigned genus, *Alloconiothyrium*, is newly described and presently represented by a single species, *A. aptrootii*, which was isolated from a soil sample in Papua New Guinea (*Verkley et al., 2014*). We therefore did not assign it to a functional guild since so little information is available. The second, *Neopestalotiopsis*, we classified as a "plant pathogen/saprotroph" based on substrates listed in species descriptions (*Maharachchikumbura et al., 2014*). The viaphyte genera of our study fit into three distinct functional guilds: *saprotroph*, *plant pathogen* and *plant pathogen/saprotroph*. Saprotroph was the dominant functional guild in terms of number of genera (70.8%; 17 out of 24) and number of isolates (82.3%, 389 out of 467). Four of the genera were classified as plant pathogens (16.7%) and three genera were classified as plant pathogen/saprotrophs (12.5%). Of the isolates, 64 were classified as plant pathogen/saprotrophs (13.7%) and fourteen were classified as plant pathogens (3.0%).

# DISCUSSION

## Viaphyte prevalence

Here, we demonstrate for the first time that a diverse array of tropical leaf endophytes can colonize woody substrates through direct contact with leaves, thus representing an ability to alternate between endophytic and saprotrophic life stages. Our results show that viaphytes are commonplace and multiple fungal species have a potential for viaphytic dispersal from within each leaf, even though it is likely that we underestimate richness due to the biases of culture-based studies (*Schmit & Lodge, 2005*) and the incompleteness of our sequencing efforts. The high frequency of viaphytic colonization suggests that the underlying mechanisms are likely mechanistically straightforward (i.e., as simple as hyphae extending from one substrate into the other), although the enzymatic potential to successfully colonize woody substrates may be taxon-dependent.

While the present viaphyte survey examined only a single tree of *N. lineatifolia*, it seems unlikely that this host is unique in allowing the transfer of endophytes to woody substrates, or that the viaphytes observed within its tissues are only able to transfer from this particular host. In other words, if the host tree and its endophytic symbionts are taken to represent what is typical for a broad-leaved tropical tree, it follows that viaphytes are likely commonplace symbionts in the leaves of tropical forests. Other studies that have demonstrated the high abundance of endophytes in tropical forests corroborate this potential (*Arnold & Lutzoni, 2007*; *Rodriguez et al., 2009*; *Thomas et al., 2016*; *Del Olmo-Ruiz & Arnold, 2017*; *Roy & Banerjee, 2018*).

Yet even if fungi with viaphytic abilities are common, the extent to which viaphytic colonization events occur in natural systems is unknown. While we placed leaves containing endophytes on sterile wood substrates, viaphytes in nature would face competition from other sources of colonization, such as spores or saprotrophs already present in the wood (*Thomas et al., 2016*). Future experiments should empirically test the ability of viaphytic fungi to successfully colonize such diverse woody substrates in the face of competition. It is likely that viaphytism and direct spore colonization each have their own set of advantages. For instance, it is possible that the carbon and water supplies inherent in leaf tissues give an advantage to viaphytic dispersal as compared to spores, especially if conditions are dry or otherwise unsuitable for spore germination. In addition, leaves could trap moisture between the leaf and substrate, and may act as barriers that exclude competing spores from being deposited on the woody substrate surfaces (*Thomas et al., 2016*). Certainly, direct spore dispersal has its own advantages in the form of reduced complexity (i.e., no intermediate colonization stage is required), increased potential travel distance via air currents (*McCartney & West, 2007*; *Calhim et al., 2018*), and much greater abundances compared to leaf-born colonies. These ideas were previously explored by (*Thomas, Vandegrift & Roy, 2020*) using a simple agent-based model. As predicted by *Thomas et al. (2016)*, in these simulations viaphytism is advantageous under adverse conditions given retention of endophyte infections and at least some trees on the landscape.

The viaphyte community of *N. lineatifolia* was characterized by a few taxa with high abundances and a large number of taxa with low abundances (Fig. 2). While this pattern is typical for culturable studies of leaf endophytes (*Arnold et al., 2000*, *2007*; *Vega et al., 2010*; *Gazis & Chaverri, 2010*; *Ikeda et al., 2014*; *Del Olmo-Ruiz & Arnold, 2017*), some patterns in the data suggest that they are partly due to methodological biases. For instance, *Penicillium* spp. and *Trichoderma* spp. were both observed to be fast growing in culture in this study, and culture-based studies are known to be biased for faster-growing taxa (*Kirk et al., 2004*). Also, given that each of the four most dominant taxa had a disproportionately high number of isolates concentrated in a single box, these dominant taxa likely colonized the sticks within their respective boxes via sporulation during the inoculation period (Fig. 3). All four of these dominant taxa readily produced a high quantity of conidia in culture. Therefore, the number of isolates for these abundant taxa should be interpreted with caution as they likely do not reflect the actual abundance in host leaves, but rather comparatively fast growth and within-box contamination. It is also

notable that our experiment did not have a true negative control, without an inoculation source, to account for true contaminants (i.e., taxa that may have originated outside of the leaves). While it is possible that some taxa detected may have been contaminants, there are several factors which suggest relatively low rates of outside contamination: (1) the thorough sterilization procedures we employed; (2) the high endophyte load in the tropics (*Arnold et al., 2000*; *Arnold & Lutzoni, 2007*); (3) the near ubiquity of detected taxa being found in tropical endophyte datasets; and (4) the restriction of common taxa to single boxes.

## Ecological strategies

It is well documented that many endophytes have a much broader host range in the endophytic state than as saprotrophs—for example, Xylariaceae, the majority of which do not typically reproduce in the litter (*Davis et al., 2003*; *Peršoh et al., 2010*; *U'Ren et al., 2016*). It is, in fact, apparently common for such endophytes to be present in the leaves of hosts upon whose wood they never fruit (*Carroll & Carroll, 1978*; *Peršoh et al., 2010*; *Unterseher, Peršoh & Schnittler, 2013*). This is evidence for a FA ecology, since latent saprotrophism is excluded as a strategy for species which are incapable of fruiting out of leaves (*Thomas et al., 2016*). It is interesting that many fungi that are not typically observed fruiting on litter, such as members of the Xylariaceae, are well known as highly competitive litter decay organisms (*Koide, Osono & Takeda, 2005*; *Osono, 2007*; *Osono et al., 2011*). It is logical that increased substrate utilization in the litter, and therefore increased resource accumulation, translates to increased ability to compete for substrates external to the litter (*Boddy, 2000*).

Latent saprotrophism is a well-documented strategy of some leaf endophytes (*Osono, 2006*; *Parfitt et al., 2010*; *Voříšková & Baldrian, 2013*). An excellent example of this ecological strategy is the fungus *Rhabdocline parkeri* (*Sherwood-Pike, Stone & Carroll, 1986*), which spends most of its lifecycle as an endophyte in the needles of *Pseudotsuga menziesii*, waiting for the needles to die (typically 4–5 years). After needle senescence, the fungus rapidly invades the surrounding needle tissues (often before they are even shed), and then produces its conidial state, followed by a small perithecial teleomorph early in the winter, soon after the leaves are shed (*Stone, 1987*). The host specificity of *R. parkerii*, and other fungi like it, is explained by the role of priority effects (*Chase, 2003*) in the latent saprotrophic habit: while priority effects may work to benefit viaphytic fungi somewhat, they serve as a strong evolutionary filter for fungi utilizing a latent saprotrophic strategy. Future studies examining viaphytic ecological strategies should focus on exploring the boundaries between viaphytic and latent saprotrophic ecologies.

## Taxonomic distribution

The viaphytes in this study belong to a wide taxonomic breadth, consisting of both Basidiomycota and Ascomycota. This implies that the benefits described by the FA hypothesis are available to members of the Basidiomycota as well, though the original idea concerned only the Ascomycota (*Carroll, 1999*). The taxonomic distribution of viaphytes from this study resemble those of general tropical leaf-endophytes described in other

work (*Arnold & Lutzoni, 2007*; *Thomas et al., 2016*; *Roy & Banerjee, 2018*). In particular, *Arnold et al. (2007)* reported a similar pattern and proportion of Eurotiomycetes, Dothideomycetes and Sordariomycetes, also noting the dominance of Ascomycota.

The wide taxonomic distribution of viaphytes suggests that viaphytic dispersal may be a deeply ancestral trait. This would parallel endophytes in general, which appear to have associated with plants since at least 400 mya (*Krings et al., 2007*). Future taxonomic and paleontological work may help inform when viaphytism emerged as a dispersal strategy within the Fungi.

## Functional guilds

Most of the viaphytic taxa in our study (17 of 24 taxa) were classified by FUNGuild as having saprotrophic abilities (Table S3). Many of these saprotrophic taxa are known wood-decay fungi, including *Xylaria* spp. and *Phanerochaete* spp. (*Nguyen et al., 2016*). In addition, our host leaves were harboring at least some species capable of physiological white-rot fungi, as evidenced by bleaching of the wood and a substantial decrease in size in several of our substrate fragments. Even some ascomyceteous molds are known to be degraders of lignin, including some *Penicillium* spp., *Trichoderma* spp., and *Fusarium oxysporum*, all of which were present among our isolates (*Rodriguez et al., 1996*; *Ryazanova, Chuprova & Luneva, 2015*). While the prevailing explanation for the occurrence of saprotrophic fungi as endophytes is that they are latent saprotrophs waiting to consume leaves upon senescence (*Peršoh, 2013*), many taxa we observed here, and others commonly isolated as endophytes, are not known to reproduce on dead leaves. Alternately, such endophytic saprotrophs may represent an evolutionary "dead-end" if they are unable to escape that state (*Bayman et al., 1998*), but our data suggests that it may be the norm for such fungi to transfer out of an endophytic state. Additionally, the presence of several taxa classified as primarily pathotrophs suggests that the facultative ability to access saprotrophic lifestyles may serve as a functional bridge for certain biotrophic species. One might expect that if biotrophs are cultivated on any given substrate, the resulting community would be dominated by fungi that were typically biotrophic, but with facultative saprotrophic abilities. This, however, is not what we find here, indicating that it is likely that a large proportion of endophytes isolated here are not transitioning to saprotrophy in a facultative manner, but as a transition back to their primary nutritional mode.

We observed several instances of fungi apparently thriving after colonizing wood. For example, despite the fact that only very few, generally host-specific, *Xylaria* are capable of fruiting from leaves (*Rogers, 2000*), *Xylaria flabelliformis* was observed fruiting directly from the woody substrates after transfer from an endophytic state. Interestingly, this taxon was found to be a common endophyte of forests in Taiwan (*Vandegrift et al., 2019*). Previously, we found five *Xylaria* species both as endophytes and as stromata on woody substrates at Los Cedros (*Thomas et al., 2016*). Emigration from leaves to wood is likely necessary for such endophytic individuals to regain reproductive potential.

## CONCLUSION

As an alternative to the latent saprotroph hypothesis, the FA hypothesis (viaphytism) suggests that many saprotrophs use endophytism to modify dispersal to their primary (i.e., reproductive) substrates (*Carroll, 1999*; *Thomas et al., 2016*; *Thomas, Vandegrift & Roy, 2020*). Here, we demonstrate for the first time that a diverse assemblage of foliar endophytes can directly colonize woody substrates from leaves, and that a high proportion of these fungi are ecological saprotrophs. This work provides new support for the FA hypothesis. While the prevalence of viaphytic dispersal in nature is currently unknown, the diversity and abundance of viaphytes observed here suggests that it may be commonplace. Viaphytic dispersal may have ramifications not only for the dispersal and competition dynamics of fungi, but also for larger scale processes, such as decomposition (*Thomas, Vandegrift & Roy, 2020*). These dynamics are largely unexplored and represent a vast potential for future research (but see, for example, *Osono (2006)*).

One such research topic that is suggested by this work concerns the effects of viaphytic dispersal on outcrossing (and thus evolutionary trajectories) of taxa utilizing this dispersal strategy. Dispersal by viaphytism could lead to an increase in outcrossing by reducing the chances of mating between spores of the same parent: spores released from the same fruiting event have a relatively high likelihood of colonizing the same nearby substrates and mating. However, if a subset of those spores delay their colonization of wood by becoming endophytes, it is likely that they increase their chances of mating with a non-sibling.

## ACKNOWLEDGEMENTS

DC Thomas aided with lab work and commented on the manuscript, H Soukup helped with sequencing. We appreciated the facilities of the field station at Reserva Los Cedros in Ecuador, where the experiment took place. Lastly, we are thankful for the thoughtful commentary on this manuscript by the editor, an anonymous reviewer, and Naupaka Zimmerman.

### Funding

Aaron Nelson received an UnderGrEBES Award, sponsored by GrEBES (the Graduate Evolutionary Biology and Ecology Students) at the University of Oregon; a McNair Scholarship and TRIO Student Support Services funding, provided by the US Department of Education; Undergraduate Research Opportunity Program funding and a Hendricks-Goodrich scholarship from the University of Oregon; a Dunbar Scholarship from the University of Oregon College of Arts and Sciences; and the Ben Selling and Andy Aitkenhead scholarships from the Oregon Office of Student Access and Completion. Roo Vandegrift was supported by a National Science Foundation Graduate Research Fellowship (DGE-0829517). The funders had no role in study design, data collection and analysis, decision to publish, or preparation of the manuscript.

## Grant Disclosures

The following grant information was disclosed by the authors:
GrEBES, University of Oregon.
US Department of Education.
University of Oregon.
University of Oregon College of Arts and Sciences.
National Science Foundation Graduate Research Fellowship: DGE-0829517.

## Competing Interests

The authors declare that they have no competing interests.

## Author Contributions

- Aaron Nelson performed the experiments, analyzed the data, prepared figures and/or tables, authored or reviewed drafts of the paper, and approved the final draft.
- Roo Vandegrift conceived and designed the experiments, performed the experiments, analyzed the data, prepared figures and/or tables, authored or reviewed drafts of the paper, and approved the final draft.
- George C. Carroll conceived and designed the experiments, authored or reviewed drafts of the paper, and approved the final draft.
- Bitty A. Roy analyzed the data, prepared figures and/or tables, authored or reviewed drafts of the paper, and approved the final draft.

## Field Study Permissions

The following information was supplied relating to field study approvals (i.e., approving body and any reference numbers):

All field work was done with the approval of the Ecuadorian Ministry of the Environment (Ministerio del Ambiente de Ecuador, Permit No. 03-2011-IC-FLO-DPAI/MA).

## DNA Deposition

The following information was supplied regarding the deposition of DNA sequences:

The ITS sequences are available at GenBank: MN421851–MN421910.

## Data Availability

Raw data and code are available at FigShare: Nelson, Aaron; Vandegrift, Roo; Carroll, George C.; A. Roy, Bitty (2019): Data from: Double Lives: Transfer of fungal endophytes from leaves to woody substrates. figshare. Dataset. DOI 10.6084/m9.figshare.9794699.v1.

## Supplemental Information

Supplemental information for this article can be found online at http://dx.doi.org/10.7717/peerj.9341#supplemental-information.

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
