# Peer review of "Double lives: transfer of fungal endophytes from leaves to woody substrates"

_PeerJ, doi:10.7717/peerj.9341_

## Round 0.1 · original submission · Major Revisions

This manuscript addresses a key topic in fungal ecology; however, the reviewers and myself have noted some methodological and statistical issues that undermine the author's conclusions as they are presented. Please see the reviewer's detailed comments and suggestions for improvement.

Reviewer 1 ·

Basic reporting

Overall, the material is well-written, and the experiment represents a novel contribution.

Experimental design

I have a few more substantial comments about the rationale for analysis and presentation of methods (see below in comments to author)

Validity of the findings

All of the data has been provided, meaningful replication was used, and the conclusions drawn from the paper are clearly linked to the Foragin Ascomycete/viaphyte hypothesis that the authors laid out in the beginning of the paper.

Additional comments

In this paper, the authors present evidence that some fungi may have a functional strategy wherein they exist inside plant leaves as endophytes in order to colonize dead wood or leaf material upon leaf senescence. The authors refer to this idea as the Foraging Ascomycete Hypothesis, or the “Viaphyte” hypothesis. Interestingly, the authors primarily focus on the possible benefits of a life history strategy for the fungal symbionts, rather than focusing on plant life history strategies or why plants may have evolved to host microbes. Overall, the material is well-written, and the experiment represents a novel contribution. I do have a few more substantial comments about the rationale for analysis and presentation of methods, in addition to some minor line comments below.

Major Comments

1) Important methodological details for molecular work and statistical analyses are not presented in several cases, as follows. L136-141 – Please provide all conditions for PCR and PCR clean-up if it was performed. L142-144 – What were the parameters or criteria you used for sequence trimming and OTU/Taxonomic grouping? L150-156 – Please describe statistical analysis methods in more details, as written no mention of linear models, random effects, or post-hoc tests used are given.
2) L147-148 – Why did you curate the UNITE database to only the genus level??? I understand that you wanted to use the FunGuild database (which requires genus to give an assignment). However, microbial taxonomic classification methods are not perfect, when you so substantially alter the database you are using as a reference for name assignment you are bound to overclassify microbial taxa when the true expected taxa are missing from the reference database. I do not believe that the desire to use FunGuild in this case warrants such a severe restriction of the reference database. FunGuild functional assignments should always be considered only as potential evidence of function. There is clear evidence in the literature of single fungal genera encompassing multiple life history strategies at the species level. (also Lines 167-8, 204-5).
3) L231-238 – This paragraph seems an important place to mention that future experiments should empirically test the ability of “viaphytic endophytes” to invade woody substrates when there are other colonizers already present. Your current experiment is fairly far removed from real conditions (sterile tongue depressors), and thus more realistic experiments are worth mentioning.

Minor Line Comments
L123 – Don’t say Roy Lab, just say were brought back to our lab at U of O for processing.
L134 – why was only a single morphotype sequenced? Can you be sure of your accuracy in morphotyping, or how frequently did you use microscopic features to assist your identification?
L170 – As written, this sentence is confusing. The fungi were isolated from wood not leaves yes? You can’t really comment on the number of taxa isolated from leaves, it is really the number of taxa found on wood that presumptively came from the same leaf group.
L185-188 – Please make it clearer what the N= stands for, is this the total number of all isolates from a box and then we take the percentage to get the final number per species, or is the N= the number for species?
Figure 1 – There are 5 isolates on the right-hand side of the figure that are noted as being cultured from 1 leaf, yet there are no isolations of these taxa? These do not appear to be the same 5 taxa mentioned in the figure legend as being unidentified, or this is confusing if they are. Please clarify on both counts to make this figure interpretable.

·

Basic reporting

Overall, this paper was well written. Figures were professionally made, the writing was well organized.

Some things that need to be fixed:

Data: I didn't see mention of raw data deposition (sequences to NCBI, tables of data in Dryad/Figshare/etc).

Citations: I was surprised to see only one of Takashi Osono's papers cited or discussed, as many of the articles he and colleagues have written are closely related to the topic of this paper. For example, see

Fukasawa, Y., T. Osono, and H. Takeda. 2009. Effects of attack of saprobic fungi on twig litter decomposition by endophytic fungi. Ecological Research 24:1067-1073

It also seems that the Discussion in particular would be strengthened by engaging more thoroughly with the existing body of work on post-senescence decompositional trajectories of fungal communities in litterfall. One that comes to mind in particular:

Voříšková J, Baldrian P. 2013. Fungal community on decomposing leaf litter undergoes rapid successional changes. ISME J. 7(3):477-86. doi: 10.1038/ismej.2012.116

Experimental design

My primary concerns with the experimental design are twofold. One is that there are no negative controls, which seem to me to be important for demonstrating that the fungi growing on the tongue depressors are not a result of contamination. The second issue is mentioned by the authors: the samples are not independent, since all samples in a given container are more likely to be similar to one another (e.g. Figure 3). Since the results are not independent, there are some elements of the results that I do not think are appropriate to report as they are currently. For example, any estimate of the mean number of taxa per leaf is likely to be spurious, since the most parsimonious explanation fo the data would be that sporulating fungi in a box contaminated other samples in that box, and so the attribution of which taxa came from which leaf fragments is at best unclear.

Validity of the findings

As mentioned above, I did not see where the raw data were deposited. It would also be useful to have the R analysis code provided. This could be either in an archival repository like FigShare/Dryad/Zenodo, or else as a supplement to this manuscript.

Given the issues with the experimental design, I think the authors need to rethink what results they present and consider what information is robust to the issues of cross contamination and the related lack of independent replication between samples.

Given the issues and as written, I don't think all of the findings are necessarily valid, but I could imagine restructuring the paper to focus on the subset of things that can still be learned from the experiment.

Additional comments

Overall, I applaud the authors for engaging (in a series of papers they have now published and some in review) with the understudied topic of the ecological roles of endophytic fungi. They have brought new ideas/thinking to the discussion that I think have a lot of merit (foraging ascomycete hypothesis, viaphytism) for explaining observations of these systems in nature. At their core, these ideas are likely to be really useful ways to think about what these organisms are doing in the hosts/ecosystems they inhabit.

My overall concern with this manuscript, however, is the emphasis on the creation of a new term, 'viaphytism'. I think the concept is useful, and I see how the previous terminology used, 'foraging Ascomycete', is not appropriate, since numerous Basiodios seem to follow the same pattern, but don't see how a new term is necessary given the existing terminology used to describe similar concepts (latent decomposers, latent saprotrophs). I wonder if the argument the authors are making, that what many endophytes are doing in transitioning between substrates of different types (live leaves -> dead woody debris) as opposed to staying put in the same substrate as it senesces (live leaves -> dead leaves) is distinct enough to warrant a new term. Many fungi found endophytically have been shown to have the ability to degrade senesced or necrotic tissue, and many of these are also known to be very cosmopolitan genera and species. In that case, it may be possible for many of these fungi to persist on/in healthy leaves, but prefer or specialize on a particular woody substrate. In other words, if the 'via' part of the term is not essential (in other words, they can be in leaves but don't have to be), then is it worth using as a description of an organism as a whole? And if it's just a temporary state, then why not just say 'saprotroph with an endophytic life stage'?

Specific comments:

Line 71: Would be good to add some citations here -- my impression is that a good number of saprophytic endophytes can at least produce conidia on fallen leaves (and thus are not trapped in a 'dead end'), even if they don't produce sexual fruiting bodies.

Lines 92-93: I think there is a distinction of kind between 'endophyte' and 'saprotroph'. The former refers to location, the second to consumption. The more appropriate comparison would be between endophytic and free-living or between biotroph and saprotroph.

Line 106: But isn't this tautological? Or are you saying in contrast to pathogenicity? Because it seems logical that pathogens of one plant existing as endophytes of another plant would also benefit in much the same way as saprotrophs might.

Line 230: Citation 'Elizabeth Arnold' is improperly formatted.

Line 251: ibid

Line 276: Typo 'viaphyptic'

Line 294: This citation is no longer in review.

Line 303: I don't get the logic here. If they can colonize wood, then aren't they saprotrophs by definition? The experimentally observed ability to grow on dead woody substrate seems to me much more convincing evidence of its ecological mode than how a taxon is described in a particular database.

Figure 1: Small point, but if there is only a single isolate for a give taxon, then it seems it should be sp. instead of spp.

---

## Round 0.2 · Minor Revisions

The reviewers and I both applaud the author's detailed response to reviews. However, while the revised manuscript is much improved there are remaining concerns about the use of the term viaphyte. The details and explanation provided in the response to reviews clarifies the term and clearly describes the rationale for its use; however, the definition of the term within the manuscript itself remains confusing and should be addressed. Please address the comments by reviewer 1 as well as comments in the attached document when making your revisions.

·

Basic reporting

No major issues remain.

Experimental design

No major issues remain.

Validity of the findings

No major issues remain.

Additional comments

I applaud the authors for engaging so constructively with the review. I appreciate the thorough response to comments and have been convinced by their argument for including the term 'viaphyte'. The discussion in the response to reviewers helped me to understand some of the subtlety that I missed the first time through. I think this is an exciting paper as much for the ideas it contains as the data it presents, and it is likely to be a valuable contribution to the field.

I did notice a number of outstanding proofreading issues, which I do not list here, including some issues with the formatting of the references (e.g. the last two are out of order and one is repeated). However I assume those will get resolved in the proof stage prior to publication (or I would certainly hope so).

---

## Round 0.3 · Minor Revisions

Thank you for your close attention to the reviewer's suggested changes to your manuscript. While the manuscript is much improved, due to the complexity and nuances of the term viaphyte it is my opinion that some additional revisions are necessary to more clearly explain the conceptual basis of this term to non-expert readers. Please see the attached document for suggested revisions to the manuscript. I look forward to receiving the revised manuscript. Thank you again.

---

## Round 0.4 · accepted · Accept

Thank you for making the suggested changes to your manuscript. I am pleased with the revisions and I feel this is a nice contribution to our understanding of endophyte biology.